# Inhaled Sedation with Volatile Anesthetics for Mechanically Ventilated Patients in Intensive Care Units: A Narrative Review

**DOI:** 10.3390/jcm12031069

**Published:** 2023-01-30

**Authors:** Khaled Ahmed Yassen, Matthieu Jabaudon, Hussah Abdullah Alsultan, Haya Almousa, Dur I Shahwar, Fatimah Yousef Alhejji, Zainab Yaseen Aljaziri

**Affiliations:** 1Anaesthesia Unit, Surgery Department, College of Medicine, King Faisal University, P.O. Box 400, Hofuf City 31982, AlAhsa, Saudi Arabia; 2Department of Perioperative Medicine, CHU Clermont-Ferand, iGReD, Universite Clermont Auvergne, CNRS, ISERM, 6300 Clermont-Ferrand, France; 3Anaesthesia Department, King Abdulaziz Hospital, P.O. Box 2477, Hofuf City 31982, AlAhsa, Saudi Arabia; 4Otolaryngology Department, AlJaber Specialized ENT and Eye Hospital, P.O. Box 36367, Hofuf City 36422, AlAhsa, Saudi Arabia; 5Family Medicine Department, AlAhsa Health Cluster, P.O. Box 5298, Hofuf City 36356, AlAhsa, Saudi Arabia

**Keywords:** volatile anesthetics, sedation, intensive care unit, mechanical ventilation, isoflurane, sevoflurane, desflurane

## Abstract

Inhaled sedation was recently approved in Europe as an alternative to intravenous sedative drugs for intensive care unit (ICU) sedation. The aim of this narrative review was to summarize the available data from the literature published between 2005 and 2023 in terms of the efficacy, safety, and potential clinical benefits of inhaled sedation for ICU mechanically ventilated patients. The results indicated that inhaled sedation reduces the time to extubation and weaning from mechanical ventilation and reduces opioid and muscle relaxant consumption, thereby possibly enhancing recovery. Several researchers have reported its potential cardio-protective, anti-inflammatory or bronchodilator properties, alongside its minimal metabolism by the liver and kidney. The reflection devices used with inhaled sedation may increase the instrumental dead space volume and could lead to hypercapnia if the ventilator settings are not optimal and the end tidal carbon dioxide is not monitored. The risk of air pollution can be prevented by the adequate scavenging of the expired gases. Minimizing atmospheric pollution can be achieved through the judicious use of the inhalation sedation for selected groups of ICU patients, where the benefits are maximized compared to intravenous sedation. Very rarely, inhaled sedation can induce malignant hyperthermia, which prompts urgent diagnosis and treatment by the ICU staff. Overall, there is growing evidence to support the benefits of inhaled sedation as an alternative for intravenous sedation in ICU mechanically ventilated patients. The indication and management of any side effects should be clearly set and protocolized by each ICU. More randomized controlled trials (RCTs) are still required to investigate whether inhaled sedation should be prioritized over the current practice of intravenous sedation.

## 1. Introduction

Inhaled sedation for critically ill patients with volatile anesthetic agents were recently revisited during the current coronavirus disease COVID-19 pandemic in view of the shortage in intravenous sedative agents. Many governments took action to address this issue by centrally managing the supply chains affected by the lockdown policies and the international travel restrictions [1,2,3,4]. Isoflurane, one of the volatile anesthetic agents, is now approved for intensive care unit (ICU) sedation in several European countries. Intravenous sedatives and their active metabolites are organ-dependent for elimination, and this can lead to unpredictable pharmacokinetics and pharmacodynamics, drug accumulation, poor clearance, and slow wake-up in critically ill patients. In contrast, volatile anesthetics are independently exhaled by the lungs and require minimal metabolism [5,6,7,8,9,10,11]. The objectives of this narrative review were to identify and discuss the published literature concerning the role of volatile anesthetics as sedatives for mechanically ventilated patients in the ICU. This review focused on lessons learned and the precautions required.

## 2. Materials and Methods

The study was approved by the local research and ethics committee (IRB KFHH No. H-05-HS-065) of King Fahad Hospital, Hofuf city, Saudi Arabia. The research approval number is RCA NO: 12-E-2021.

In this narrative review, databases including Embase, CINAHL, Scopus, Google Scholar, Google, Science Direct, ProQuest, ISI Web of Knowledge, and PubMed were searched to obtain the related literature published in the English language. The key words were: Inhalation Sedation; volatile anesthetic agents; Intensive Care Unit; Mechanical ventilation; Isoflurane; Sevoflurane; and Desflurane.

Study selection: Randomized controlled trials (RCT), observational studies, retrospective studies, and reviews were included. The patients were adult humans (age >18 years) under mechanical ventilation in medical, surgical, and specialized cardiac ICUs, but not neurosurgical ICUs. The recorded data included the volatile anesthesia sedation effects on ventilation and weaning, hemodynamics stability, organ dysfunction, cognitive function, recovery, and the risk of air pollution.

## 3. Results

The literature search focused on the period between 2005 and 2023. Seventy-four peer-reviewed studies were selected as a result of the initial reading of the abstracts and reviewing the full text for each article. Sixty-seven peer-reviewed studies were included that identified and covered different aspects of inhalation sedation practice. Six recent and different studies summarized the various practical aspects of concern with inhaled sedation among COVID-19 patients during the last pandemic. (Table 1).

Inhaled sedation reduced the extubation and weaning times of mechanical ventilation, lowered opioids (analgesic sparing effect) and muscle relaxants consumption, enhanced recovery and minimized delirium. Table 2 demonstrates the published controlled trials that demonstrated the analgesic sparing effect of inhalation sedation compared to the other, traditional intravenous sedatives. The improvement in the quality of recovery was demonstrated by Ostermann and his colleagues in their systematic review, Mesnil et al. in their clinical trial, and Blondonnet et al. in their national survey [12,13,14]. Inhaled sedation must utilize anesthesia reflection devices (ACD-s), such as the Sedaconda Anesthetic Conserving device (Sedaconda-ACD. Sedana Medical, Danderyd, Sweden) and the MIRUS system (Pall Medical, Dreieich, Germany). An Illustration of ICU setup for inhalation sedation presented in Figure 1. These devices significantly increase the dead space and require tidal volumes greater than 350 mL and increasing the respiratory rate of the ventilators to avoid hypercapnia, together with the monitoring of any potential auto-positive end expiratory pressure (PEEP) [15,16,17]. This led to the development of smaller reflection devices, with only 50 mL dead space (ACD-S), in 2017. These ACD-s were not associated with hypercapnia with the tidal volumes > 200 mL [18]. Table 2 presents two randomized controlled trials and one prospective study demonstrating the ability of analgesic drugs’ sparing effects of the inhaled sedation [8,19,20]. Table 3 summarizes the controlled trials that support the role of volatile agents in preserving systemic hemodynamics compared to intravenous propofol [7,21,22]. Fifteen studies also addressed the organ-protective properties of these volatile anesthetic agents. Other studies addressed different topics and will be discussed in sequence.

**Table 1 jcm-12-01069-t001:** Published studies in chronological on sedating COVID-19 mechanically ventilated patients with volatile anesthetic agents.

Study	Study Type and Population	Sedative Agents	Conclusion
Flinspach A et al. [12] (2020)	Retrospective analysis of five COVID-19 patients admitted to the ICU and requiring mechanical ventilation	Isoflurane	Feasibility of inhaled sedation in ICU and patients undergoing ECMO. Adequate sedation to facilitate ventilator synchrony, prone positioning
Kermad et al. [13] (2021)	Retrospective study included 20 patients with COVID-19 ARDS admitted to the ICU	Isoflurane as inhalational and propofol as intravenous sedative	Isoflurane provides sufficiently deep sedation with less polypharmacy, less NMBA use and lower opioid doses.
Nieuwenhuijs-Moeke et al. [14] (2020)	Editorial	Sevoflurane	Feasibility of volatiles anesthetics and their potential beneficial effects of volatile anesthetics on systemic inflammation, sepsis, and ARDS in mechanically ventilated COVID-19 patients
Suleiman A et al. [15] (2021)	Review of literature	Isoflurane, Sevoflurane and Desflurane	Short-term sedation with volatile anesthetics may be beneficial in severe stages of COVID-19 ARDS. They have proven benefits at the molecular, cellular, and tissue levels.
Kaura and Hopkins [16] (2020)	Editorial	Sevoflurane	Theoretical risk of MH among COVID-19 and educating ICU staff to manage MH
Bellgardt et al. [17] (2021)	Review of Critically ill COVID-19 patients undergoing ECMO	Isoflurane	Benefit of spontaneous breathing and deep sedation in prone position.

ARDS: acute respiratory distress syndrome, COVID-19: coronavirus disease, ECMO: Extracorporeal membrane oxygenation, ICU: intensive care unit, IV: intravenous, MH: malignant hyperthermia, NMBA: neuromuscular blocker agents.

**Table 2 jcm-12-01069-t002:** Published trials in a chronological order demonstrating the analgesic drugs sparing effect of inhaled sedation compared to intravenous sedation.

Study	Study Type and Population	Inhaled Sedation Group (Drug, *n*)	Intravenous Sedation Group (Drug, *n*)	Mean Sedation Duration	Target Sedation Level	Outcomes
Jung S. et al. [8] (2020)	Prospective study of Patients scheduled for elective head and neck surgery with tracheostomy (post-operative ICU sedation)	Sevoflurane (*n* = 25)	Propofol (*n* = 24)	Inhaled: 771.0 ± 388.4 min Propofol: 1508.2 ± 2074.7 min	RASS −2 to −3 CPOT < 3	Post-operative opioid consumption. Monitored the proper initial end-tidal concentration of Sevoflurane in patients underwent head and neck surgery with tracheostomy
Mesnil et al. [19] (2011)	Randomized controlled trial included ICU patients who need more than 24 h of sedation	Sevoflurane (*n* = 19)	Propofol (*n* = 14) Midazolam (*n* = 14)	Inhaled: 50 h Propofol: 57 h Midazolam: 50 h	RSS 3–4	Awakening and extubation time, RSS monitored, post extubation opioid consumption, post extubation hallucination, renal and hepatic function.
Meiser A et al. [21] (2005)	Randomized, controlled of Adult ICU patients who are expected to need at least 24 h of sedation	Isoflurane (*n* = 146)	Propofol (*n* = 146)	Inhaled: 48 h Propofol: 48 h	RASS −1 to −4	RASS, adverse events monitored, opioid consumption, ventilation setting and awakening and extubation times monitored

ICU: intensive care units, RSS: The Ramsay Sedation Scale, RASS: Richmond Agitation Sedation Scale.

**Table 3 jcm-12-01069-t003:** Published randomized controlled trials in a chronological order demonstrating the inhaled sedation preserving effect on the hemodynamics as compared to intravenous propofol sedation.

Study	Study Type and Population	Inhaled Sedation Group (Drug, *n*)	Intravenous Sedation Group (Drug, *n*)	Observation Duration	Target Sedation Level	Included Outcomes
Yassen K et al. [7] (2016)	Prospective randomized hospital based comparative study of Liver transplant adult patients planned for weaning of mechanical ventilation	Desflurane (*n* = 30)	Propanol (*n* = 30)	Inhaled:6 h Propofol:8 h	PSI 50–75/h	HR, MAP, CO, SVR, Fetch, consumption of vasoactive drugs, Fentanyl requirements, estuation time, psychometric tests, and total cost.
Migliari, M. [22] (2009)	Randomized controlled trail of hemodynamically stable adult ICU patients requiring sedation for Mechanical Ventilation	Sevoflurane (*n* = 17)	Propofol and Remifentanil (*n* = 17)	Inhaled Phase: 2 h Propofol and Remifentanil Phase: 2 h	Ramsay score ≥ 4 and a RASS ≤ −3	Vt, RR, MV, Fio2, Et CO2, HR, IBP, CVP, SpO2, and internal body temperature, C rs, R aw, PEEPi, time to action /awake and Ambient contamination from sevoflurane.
Miser A et al. [23] (2021)	Randomized, controlled trail of Adult ICU patients who are expected to need at least 24 h of sedation	Isoflurane (*n* = 146)	Propanol (*n* = 146)	Inhaled: 48 h Propanol: 48 h	RASS −1 to −4	RASS, adverse (hemodynamic), events, opioid consumption, ventilation setting and awakening and estuation time.
Souk up et al. [24] (2023)	Prospective, randomized-controlled phase-Ibis monocentric clinical-trial	Sevoflurane (*n* = 39)	Propofol (*n* = 40)	Inhaled >48 h Propofol: 48 h	RASS −1 to −4	RASS, hemodynamics, opioid consumption, ventilation, extubation time, Length of hospital stay

ICU: intensive care unit, RASS: Richmond agitation sedation scale, VT: Tidal Volume, RR: Respiratory Rate, MV: Minute Ventilation, FiO2: inspiratory oxygen fraction, Et CO2: End Tidal Co2, HR: Heart Rate, IBP: invasive arterial blood pressure, CVP: central venous pressure, SpO2: peripheral oxygen saturation, C rs: respiratory system compliance, R aw: airway resistance, PEEPi: intrinsic PEEP, PSI: Patient Sate Index, MAP: Mean Arterial Pressure, CO: Cardiac Output, SVR: systemic vascular resistance, FTc: corrected flow time.

**Figure 1 jcm-12-01069-f001:**
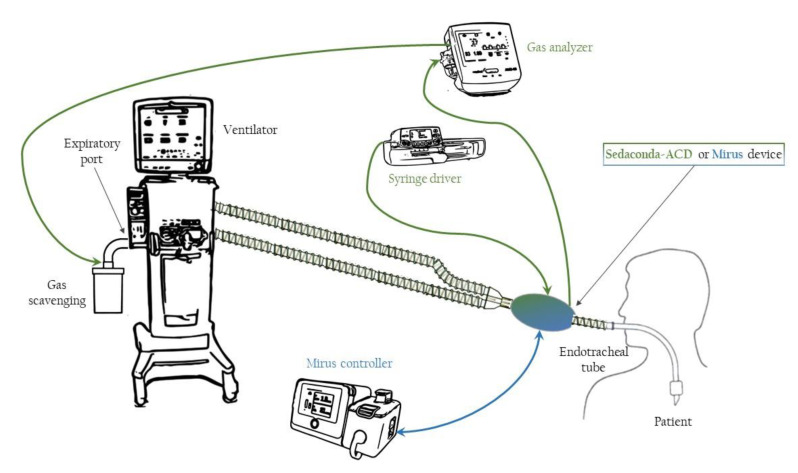
An Illustration of ICU setup for inhalation sedation. Reproduced with permission from Jabaudon M, et al. Anaesth Crit Care Pain Med. 2022. [25].

## 4. Discussion

Inhaled sedation is an effective and safe alternative to intravenous sedation in ICUs, as reported by Bisbal et al. in their prospective observational study (2011) [26]. However, further discussion is needed to guide towards the best clinical practice and the selection of appropriate patients for inhaled sedation. 

### 4.1. Sedating COVID-19 Patients

The shortage in the supply of intravenous sedatives during the COVID-19 pandemic led to the need for an alternative method of sedation. The use of inhalational agents was explored by several intensivists. In a case series and a systematic analysis by Flinspach A et al. in 2020, they noted that sedation by volatile anesthetic agents leads to deep sedation in critically ill COVID-19 patients [12]. This facilitated the mechanical ventilation during the prone position and improved the degree of synchronization with ICU ventilators. Deep sedation reduced the aerosol generation associated with coughing and decreased the inadvertent extubation. Kermad and his colleagues, in their retrospective chart review (2021), reported that the COVID-19 patients were adequately sedated with isoflurane and, hence, consumed less neuromuscular blocking agents and opioids compared to those sedated with propofol. However, in the severe forms of COVID-19, higher sedative doses of both isoflurane and propofol were required [13].

In 2020, the Nieuwenhuijs-Moeke et al. editorial presented other beneficial effects of inhaled sedation among mechanically ventilated COVID-19 patients suffering from inflammation and sepsis [14]. A recent literature review by Suleiman A et al. (2021) suggested that short-term sedation with volatile anesthetics could be beneficial with severe COVID-19 ARDS based on the molecular, cellular and tissue evidence [15].

One important concern was raised by Kaura and Hopkins about the possibility to induce malignant hyperthermia (MH) among COVID-19 or non-COVID-19 patients, if applied in a wider scale. They emphasized the need to take this into consideration and recommended the need to educate ICU staff about the methods of the diagnosis and management of MH [16]. 

On the other hand, a review by Bellgardt and his colleagues (2021) focused on the technical details of administrating volatile anesthetic agents to COVID-19 patients on extracorporeal membrane oxygenation (ECMO) [17]. 

### 4.2. Analgesic Drugs Sparing Effect 

Critically ill patients in need of mechanical ventilation are particularly challenging because they require polypharmacy. Inhaled sedation can lower the requirement of analgesic drugs, as reported by Meiser A et al. in their RCT (2005) [21] and in a review (2012) by Misra et al. [27]. Both reported a notable reduction in the consumption of opioids with sevoflurane and isoflurane sedation when compared to intravenous propofol. The RCT of Mesnil et al. (2011) also observed a reduction in post-extubation morphine consumption among critical ill patients sedated with sevoflurane vs. propofol or midazolam [19]. Mo et al., in their systematic review (2019), provided evidence that inhalational sedation was not inferior to other standard intravenous sedatives regarding pain relief [28].

Lower remifentanil consumption with sevoflurane sedation was also reported in comparison to propofol by Jung, S. et al. in their prospective study (2020) [8]. These findings can be explained by the ability of inhaled sedatives to block the N-methyl-D-aspartate receptors (antagonist activity). However, in a recently published prospective, randomized-controlled phase-IIb monocentric clinical-trial by Soukup et al. (2023), sevoflurane sedation (>48 h) compared to propofol had a lower opioid requirement of remifentanil (400 μg/h vs. 500 μg/h, *p* = 0.007) and of sufentanil 40 μg/h vs. 30 μg/h, *p* = 0.007) [28]. More RCTs are still required (Table 2).

### 4.3. Preserved Systemic Hemodynamics 

Migliari et al. reported a significant increase in heart rate with sevoflurane vs. propofol, despite comparable arterial and central venous pressures [22]. A recently published randomized, controlled trial (RCT) in 2021, by Meiser et al., demonstrated how Isoflurane sedation was not inferior to propofol and with no significant differences in hemodynamics [23]. Desflurane is rarely used as an inhalation sedative. In a prospective randomized hospital based comparative study (2016) by Yassen et al., they investigated the postoperative sedation with desflurane vs. propofol among mechanically ventilated liver transplant recipients and reported the beneficial effects on systemic vascular resistance and mean arterial blood pressure with Desflurane sedation [7]. 

Recently, Soukup et al. (2023) compared sevoflurane sedation to propofol and found that the hemodynamics were not different [28] (Table 3).

### 4.4. Organ-Protective Properties

A reduction in the need for inotropic support following coronary bypass graft surgery with sevoflurane vs. propofol was mentioned in Soukup et al.’s review in 2009, Steurer et al. (RCT) in 2012 and Soro et al. (RCT) in 2012 [29,30,31]. A significant reduction in troponin T concentrations were also reported by Steurer et al. [29]. A review conducted by Orriach et al. (2013) reported that sevoflurane postoperative sedation reduced oxygen consumption and lowered the troponin I concentrations in the blood levels [32]. However, the RCTs by Flier et al. in 2010 [33] and Wasowicz M et al. in 2018 [34] only detected limited evidence of cardiac protection. Soukup et al. [29], Steurer et al. [30] and others did observe these organ protective effects in other organs, such as the brain, lung, liver and bowels in their studies published between 2003 and 2014 [35,36,37,38,39,40]. In reviews by Jerath et al. and O’Gara et al. in 2016 and an RCT by Jabaudon et al. in 2017, attributed these organ protective properties to the anti-inflammatory effects of the volatile anesthetic agents and the reduced production of pro-inflammatory markers and cytokines [41,42,43].

### 4.5. Potential Effects on Respiratory Functions

Volatile anesthetics can benefit injured alveoli and improve arterial oxygenation with their anti-inflammatory properties, as demonstrated by several researchers, particularly in patients with acute respiratory distress syndrome (ARDS). Steurer, M. et al., in their randomized controlled trial (2012), reported an improvement in the oxygenation index with sevoflurane sedation following cardiac surgery compared with propofol [30]. In a prospective cross-over study (2009) by Migliari et al., an increase in arterial carbon dioxide tension (PaCO_2_) was noted, which was only resolved by increasing the tidal volumes [22].

Meiser et al. [21] and Krannich et al. [44] both confirmed these beneficial respiratory effects of the volatile anesthetics, but also recommended monitoring the PaCO_2_ levels frequently. Jabaudon et al., in their RCT among patients with ARDS, also observed that sevoflurane improved oxygenation and decreased epithelial injury when compared with midazolam sedation.

Furthermore, Ruszkai et al., in their case report (2014), reported the successful management of a patient suffering from an acute attack of bronchial asthma with inhaled sedation [45]. Blondonnet et al. designed a RCT and named it the SESAR trial (Sevoflurane for Sedation in ARDS), which is currently investigating the efficacy of sevoflurane compared to propofol, but the results are not yet available [46].

Finally, the effect of inhalation agents on the hypoxic pulmonary vasoconstriction (HPV) response has been discussed by several reviewers. HPV physiological redirects the blood flow from the non-ventilated hypoxic areas of the lung to other ventilated lung alveoli, this helps to limit intrapulmonary shunting and optimize the ventilation/perfusion (V/Q) ratio, which then minimizes the fall in arterial oxygen pressure (PaO_2_). Volatile anesthetics, in a dose-dependent manner, can attenuate the HPV response far more than intravenous sedatives. However, the administration of volatile anesthetics, between 0.5 to 1.5 MAC, demonstrated only a mild effect because of their compensatory bronchodilator and anti-inflammatory effects [47,48,49].

### 4.6. Renal Function under Inhaled Sedation

In a double-blinded, placebo-controlled, multicenter study (2003), Julier K et al. demonstrated the sevoflurane renal protective effect among cardiac patients following cardiac bypass surgery [35]. Röhm et al. also noted, in a prospective, randomized, single-blinded study (2009), that the renal integrity remained unchanged, despite the increase in the inorganic fluoride blood levels [50]. Sedation with volatile anesthetics is characterized by rapid pulmonary elimination which makes it suitable for patients with hepatic or renal failure [51]. In a prospective controlled study (2014) by Perbet et al., they reported that, despite the increase in the plasma fluoride levels during the 48-h inhaled sedation period, they observed no signs of nephrotoxicity [52]. Mesnil et al., Meiser et al. and Jabaudon et al., and all reported no adverse effects with sevoflurane sedation [19,21,43]. In contrast, intravenous anesthetic/sedative agents depend on end-organ elimination, and this leads to unpredictable pharmacokinetics, pharmacodynamics, and adverse outcomes. This can range between delirium and life-threatening propofol infusion syndrome.

However, inhaled sedation is not without precautions; Muyldermans et al. described a case in 2016 of partial nephrogenic diabetes insipidus that developed in a burned patient following prolonged sedation with low expired fractions of sevoflurane [53]. They advised that anesthesiologists and intensivists should always be aware of this rare incidence that can develop with sevoflurane, whether used for general anesthesia or for inhaled sedation, particularly following prolonged surgery or sedation. L’Heudé et al. also described similar findings in their retrospective study (2019). They encountered the rare development of nephrogenic diabetes insipidus (NDI) with the prolonged exposure to high-doses of sevoflurane. They recommended that these clinical findings need to be better investigated via future prospective studies [54]. Jerath A et al. demonstrated, in a pilot randomized controlled trial (2020), that isoflurane sedation increases the levels of serum fluoride concentrations in the blood, but without any significant reduction in the renal functions [55].

### 4.7. Enhancing Recovery and Cognitive Functions

In a small randomized controlled trial, Mesnil et al. demonstrated that long-term sedation with inhaled sevoflurane, compared to propofol, did reduce the wake-up time, extubation time and post-extubation morphine consumption [19]. Their results are in accordance with the studies by Meiser et al., Soukup et al., and Sackey et al. [19,29,56] and supported by Jerath et al. and Landoni et al.’s systematic reviews and meta-analysis [10,57]. However, no significant differences could be detected in terms of the hospital/ICU durations of stay. Hellström et al., in another randomized controlled clinical trial in 2012, confirmed that a significant reduction in the wake-up time post-cardiac surgery was observed with sevoflurane sedation vs. propofol [58]. Foudraine et al. added, in an observational propensity score-matched study (2021), that the delirium incidence was reduced among post-cardiac arrest patients when sevoflurane sedation was combined with targeted temperature management [59]. In another study, by Hanafy et al., that included 24 post-cardiac surgical patients, they found that the time to be extubated with isoflurane sedation was significantly shorter compared to midazolam sedation [60].

### 4.8. Air Pollution Risk

Strategies to minimize the individual exposure to air pollution with inhaled sedatives vary between countries. Exposure to low concentrations of volatile anesthetic agents, as measured by passive dosimeter scan, can only be achieved through improvements in hospital ventilation, scavenging systems and reducing exposure time to less than 8 h. Many countries do not have a time limit for exposure. Herzog-Misery et al. (2018) reviewed the effects on occupational health and presented strategies to minimize exposure and pollution [61]. Sackey et al. reported, in a prospective observational study (2005), that isoflurane levels in the air were less than the recommended international exposure limits when AnaConDa was in use [62]. Both Herzog-Niescery J et al. and Sackey et al., in their observational studies, recommended that an effective air conditioner, with at least 6–8 air changes per hour, is essential if room pollution is to be minimized. Migliari et al. and Accorsi et al. stated that sevoflurane concentrations in the air should not exceed the limit defined by the National Institute for Occupational Safety and Health in USA (2 ppm). Few studies looked into and identified the medical side effects that can develop from the prolonged exposure to the inhaled anesthetic agents [22,63].

The air-pollution and global warming risks from volatile anesthetics are well known. The fact that inhaled anesthetics are greenhouse gases and/or ozone-depleting agents emphasizes their contribution towards global warming.

Inhalation anesthesia agents undergo minimal metabolism and are eliminated, unchanged, as waste gas with each exhalation. This leads to the pollution of the environment and lasts for years. The air global warming effects of these gases have variable atmospheric lifetimes depending on their carbon dioxide content. Varughese and Ahmed, in their narrative review (2021), discussed the measures necessary to minimize the impact on the environment by these volatile anesthetic agents [64]. They advised using well-maintained ventilation and scavenging systems, coupled with the monitoring of the environmental air concentrations, for these anesthesia agents.

The process of the handling and preparation of inhalation devices are considered sources for volatile agent leakage, as reported by Alexander et al. in 2017. It is crucial to consider the long-term and cumulative effects of the volatile agents in order to pursue strategies to mitigate the associated risks to the environment.

Environmental pollution and the effect on the ozone layer by specific volatile anesthetics have a prolonged, lifetime effect. Specific volatile agents, such as Desflurane and N_2_O, need to be reduced. The carbon dioxide equivalent of Desflurane is higher than that of sevoflurane and isoflurane, which makes Desflurane not environment friendly [65,66].

The judicious use of the volatile anesthetics should only be indicated to selected groups of ICU patients, where the benefits outweigh the risks.

The use of activated charcoal filters in the expiratory limb of the ventilator cycle will help to reduce the degree of air pollution in the surroundings, but this method has its own limitation and cost. ICU ventilators still require high oxygen and air mixture flows, which mandates an increase in the volatile agent inspired percentage in order to stabilize the inhaled concentration.

Another important recommendation to reduce leaks and environment pollution is the adoption of a routine maintenance program that checks all of the equipment involved in the process of inhaled sedation, this includes the scavenging system. Training and educating both the operating rooms and ICU staff is also necessary. Herzog-Niescery J et al. recommended that extra care is required with MIRUS, particularly during the process of refilling, to reduce leaks and, hence, air pollution [67].

### 4.9. Is Sedation Depth Monitoring Essential?

Sedation depth varies one from patient to another. Individual variations can affect the systemic hemodynamic, weaning and recovery. Orriach et al. (2013) and Jabaudon et al. (2017) monitored sedation with the bispectral index (BIS) [32,43]. Romagnoli et al. adopted the Richmond Agitation Sedation Scale (RASS), and they reported that the minimum alveolar concentration (MAC) of the inhaled agents correlates negatively with the RASS values [68]. Nitzschke et al. demonstrated, in their prospective, controlled, sequential two-arm clinical study (2014), that BIS-guided sedation significantly reduced the sevoflurane plasma concentration and rescue doses of noradrenaline during on-pump cardiac surgery [69]. In a RCT (2015) by Sayed et al., the monitoring sedation depth among liver transplant recipients, post-operatively, with the patient state index (PSI) helped to preserve the systemic hemodynamics and enhanced recovery. The total consumed doses of sedatives guided by PSI were less than those guided by RASS [70]. In a prospective interventional study (2020), Blanchard et al. demonstrated that the minimum alveolar concentration of inhaled anesthetic agents (MAC) can be used as a sedative depth monitor. The increase in MAC was in correlation with the decrease in the RASS values (r = −0.83, *p* < 0.001). Monitoring the sedation depth preserves the spontaneous breathing activity and reduces the need for muscle relaxants [71].

Applying the consensual/international recommendations on sedation is necessary, and prioritizing the use of a score for sedation as a regular assessment tool is recommended to reduce delirium and prevent ventilator patient dys-synchrony [72]. The authors of this current review believe in the multi-modal monitoring approach and the respect of individual variations.

### 4.10. Which Inhalation Agent(s) Should Be Used?

Isoflurane was reported by several experienced research groups as their preferred inhalation agent. Isoflurane is cheaper and more potent than sevoflurane. The potency of Isoflurane allows the consumption of lower drug volumes, reduces the cost, and is currently approved for the sedation of ICU patients in Europe [23]. Inhalation anesthetics are metabolized at varying degrees. Sevoflurane, Isoflurane and Desflurane are metabolized by the liver, at rates of 2–5%, 0.2% and 0.02%, respectively. Sevoflurane has been associated with some cases of reversible polyuria when used at high doses and for prolonged durations. Desflurane requires pressurized vaporizers. The MIRUS device can provide inhalation sedation with isoflurane, sevoflurane and desflurane. However, Sedaconda can only be used with Isoflurane and Sevoflurane [23,56].

### 4.11. Which Patients Could Be the Best Candidates for Inhaled Sedation?

Patients receiving intravenous sedatives for prolonged durations may benefit from an alternative sedative technique, such as inhaled sedation, which is characterized by a very low metabolism. This approach would allow time for the accumulated intravenous sedatives and opioids to be metabolized and cleared. Critically ill patients under mechanical ventilation can benefit from inhaled sedation to facilitate weaning, as it reduces the requirement of opioids and neuromuscular drugs. Elderly patients may also benefit from volatile anesthetics as they can preserve the cognitive functions and reduce the incidence of delirium compared to other intravenous sedatives. Post-cardiac arrest patients are considered good candidates. Foudraine et al. reported decreased delirium and reduced hospital stay when sevoflurane sedation was combined with a targeted temperature management in post-cardiac arrest patients [59]. In a retrospective study from Hellstrom et al. that included 12 post-cardiac arrest patients, isoflurane sedation allowed for early neurologic assessment [73]. Krannich et al. (2017) also suggest that inhaled sedation could specifically benefit cardiac arrest survivors [44].

Patients who require deep sedation to facilitate ventilator synchronization in prone positioning, such as critically ill COVID-19 patients, are good candidates for inhaled sedation [12,17,74]. However, Becher T et al. conducted a multicenter randomized controlled trial (2022) and observed less pronounced improvements in the oxygenation index and V/Q mismatching in patients with (ARDS) and acute hypoxemic respiratory failure (AHRF) in the isoflurane group compared to IV sedation with propofol. The authors attributed this difference in the results from the previous studies to the fact that Isoflurane has fewer organ-protective properties than Sevoflurane [75].

### 4.12. Prolonged Inhalation Sedation Pros and Cons?

Critically ill patients who are expected to be sedated for a prolonged duration on mechanical ventilators will require daily wake up trials in order to reduce the duration of ventilation and enhance the return of spontaneous breathing. In their clinical trial (2004), Sackey et al. found that prolonged isoflurane sedation >12 h was safe and possessed a low risk of drug accumulation compared to other intravenous sedatives [76]. Inhaled anesthetics had also been recommended by Redaelli et al. in 2013 for the sedation of ventilator-dependent ICU patients and for those with drug abuse or with severe ARDS in need for long periods of sedation and paralysis [77]. Gallego et al., in a comparative study (2014), looked into the renal and hepatic integrity following long-term sevoflurane sedation among animals. They came to the conclusion that neither sevoflurane nor propofol had any negative effects on the renal or hepatic functions following prolonged exposure [78].

Prolonged sedation (>24 h) with inhaled sevoflurane was reported by Mesnil et al., in their (RCT) (2011), as an effective alternative for propofol or midazolam. They also noted an enhanced recovery and reduced analgesics consumption with sevoflurane [19].

In another recently published RCT (2023), by Soukup and his colleagues, the sedated ICU patients in their trial with sevoflurane (>48 h) required less opioids and less time to breath spontaneously when compared to the propofol-based sedation regime. Based on this RCT, sevoflurane could be considered safe for long-term sedation and non-inferior to propofol [28].

The risk–benefit analysis is always important prior to initiating an alternative method for sedation. The refractory status epilepticus and status asthmaticus are two examples of short and early clinical indications for inhaled sedation, where the benefits outweigh the risks. However, this should not be the case with elderly, critically ill patients or patients with brain trauma who are highly susceptible to neurotoxicity. The theoretical concern of compound A and fluoride nephrotoxicity was investigated and could be linked to prolonged sevoflurane sedation. Inhalation sedation as an alternative method, if practiced in a judicious way, should be associated with reduced cognitive dysfunction (POCD). More research in this field is still required [79,80].

### 4.13. Practical Aspects and Clinical Application

The lack of familiarity in practicing inhaled sedation is a major obstacle that needs to be addressed. Training and awareness about the benefits of inhaled sedation as an alternative to intravenous sedation among intensivists represents the main challenge, particularly in countries where the intensive care training track is separated from the anesthesia training track. Furthermore, sufficient studies of a RCT nature are required to compare inhalation sedation to intravenous sedation. Other obstacles facing the application logistics of volatile anesthesia need to be addressed inside each hospital.

### 4.14. What Are the Other Obstacles and Precautions with Inhaled Sedation Practice?

ICU vaporizer SedaConda-ACD is cheaper than the MIRUS, but requires additional monitors for the inhaled and expired anesthetic agent’s concentrations. This is usually not available in ordinary ICU settings and varies between countries.

ARDs patients with severe pulmonary disease require high minute volumes, which could be a challenge during inhalation sedation. Increasing the tidal volumes may take care of the dead space effects of volatile agent devices; however, in ARDS, such as patients who suffer from high dead spaces as a result of a very high VQ mismatch, this might be difficult. In this situation, volatiles should be delivered at high minute ventilation (15–25 L/min) by an anesthesia machine or by an ICU ventilator that is coupled with a miniature vaporizer. The increase in the minute ventilation requires that the volatile anesthetics infusion rate needs to be increased hand to hand in order to keep the volatile agents’ end tidal concentrations constant [81,82,83].

Finally, the availability and cost of ACD in many countries around the world, particularly in developing countries, remains a challenge that needs to be addressed by the manufactures.

## 5. Conclusions

There is growing evidence from the published literature in support of volatile anesthetic agents as sedatives for mechanically ventilated ICU patients, but in specific clinical situations. The refractory status epilepticus, status asthmaticus and patients with hepatic or renal impairment are among those specific indications, as the benefits outweigh the risks. Several studies have demonstrated their cardio-protective, anti-inflammatory and bronchodilator properties, alongside their minimal metabolism. Inhaled sedation reduces the time of mechanical ventilation and lowers opioid consumption, as well as enhancing recovery and weaning from ventilators. Providing intermittent periods of inhaled sedation as an alternative to a prolonged intravenous sedative regime will provide time for the body to metabolize and excrete the accumulated intravenous sedatives and their active metabolites.

Indications for inhaled sedation should be set upon by each ICU team, according to the patient’s needs and the training of the staff. More clinical RCTs to compare inhalation sedation to intravenous sedation are needed. Developing strict measures to reduce the effect of volatile anesthetics on the environment is essential to avoid any harm.

## Data Availability

A narrative review with references retrieved from databases of Embase, CINAHL, Scopus, Google Scholar, Google, Science Direct, ProQuest, ISI Web of Knowledge, and PubMed were searched to obtain the related literature published in the English language.

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
