# Peer review of "Inhaled Sedation with Volatile Anesthetics for Mechanically Ventilated Patients in Intensive Care Units: A Narrative Review"

_jcm, 2023, doi:10.3390/jcm12031069_

Round 1

Reviewer 1 Report

The authors' idea of taking stock regards the volatile anesthetics in the ICU as an alternative to intravenous sedation is relevant and helpful to intensivists. Especially considering their increasing use during the Covid-19 pandemic, and, consequently, the new data emerged. However, this narrative review is often unclear, there are overlaps and overall is difficult to read.

The division of the paper into paragraphs should make for well-organized reading, but this is not currently the case.

For each paragraph, a brief introduction should be written explaining why it is crucial, what is known from the literature reviewed, and what is not yet understood.

The content of each paragraph should adhere closely to the topic under consideration. The results reported in each paragraph must relate to the specific theme, with order and without repetitions that should be carefully avoided.

It might be helpful to include in the text the publication date of the studies (there is only sometimes) and to give them a chronological order since the time interval covered by the review is wide. References to renal effects are dating, which should be pointed out.

Moreover, it should always be written whether the reference is about a clinical study or a review.

Adverse events should be collected in a separate paragraph.

Also, the training needs of professionals, and the practical aspects of clinical application, should be collected in a separate paragraph.

Tables should be revised in this logic.

Author Response

Point 1

For each paragraph, a brief introduction should be written explaining why it is crucial, what is known from the literature reviewed, and what is not yet understood.

Response 1:

Thank you for the comment, we agree. The necessary adjustment were made in each major paragraph

Point 2

The content of each paragraph should adhere closely to the topic under consideration. The results reported in each paragraph must relate to the specific theme, with order and without repetitions that should be carefully avoided. It might be helpful to include in the text the publication date of the studies (there is only sometimes) and to give them a chronological order since the time interval covered by the review is wide.

Response 2:

The chronological order according to the publication dates was rearranged in all tables and throughout the text with year of publication added between brackets, references were updated and revised

Point 3

References to renal effects are dating, which should be pointed out.

Response 3: New references added and arranged according to the chronological order of publication dates page 9 and 10 in color highlighted

Point 4

It should always be written whether the reference is about a clinical study or a review.

Response 4:

Thank you for highlighting this point. We agree with it and changes were made to mention the type of the studies throughout the text.  Each reference was revised and identified if clinical study or review and added to text. Tables were revised and the type of each study was identified and added.

Point 5

Adverse events should be collected in a separate paragraph.

Response 5: Addressed under a paragraph titled Pros and Cons

Also addressed under title Obstacles and precautions.

Pages 10-12

Point 6

The training needs of professionals, and the practical aspects of clinical application, should be collected in a separate paragraph.

Response 6: A new paragraph was added to address the issue of training and practical aspects and difficulties that face the clinical application of inhalation sedation. Page 11

Point 7

Tables should be revised in this logic.

Response 7: All tables revised and each reference study type identified and rearranged in chronological order.

Reviewer 2 Report

Good review of inhalaltional sedation use in ICU .

There have been a number of review articles in different journals on this topic

Inhaled sedation in the intensive care unit. Jabaudon M, Zhai R, Blondonnet R, Bonda WLM. Anaesth Crit Care Pain Med. 2022 Oct;41(5):101133

Some comments  for authors

The review only describes the advantages as  feasible to give inhalational sedation and decreasing the dosages of muscle relaxants as well as sedative – which is not unexpected.

Harmful effects of prolonged inhalational agents need to be highlighted more .

Effect of inhalational anaesthetic agents on hypoxic pulmonary vasoconstriction – one of the compensatory mechanism to minimise hypoxia in patients with pulmonary  shunt (as happens in ARDS) also need to be highlighted.

Role of waste anaesthetic gases on environmental pollution not highlighted.

Authors have dealt with only ICU pollution and gas scavenging. The waste anaesthetic gas effect on carbon load as well as greenhouse effect needs to be highlighted. These effects are forcing anaesthetist to stop using inhalational agents in operating room for anaesthesia.

So this may not have generalised use in ICUs.

Presently it may have selective role in patients requiring very high IV sedatives and also for giving gap of IV drug free time. It can be use full as supplement to high iv sedative dosages.

There is need to have more controlled studies  comparing two types of sedation for more conclusive evidence.

Adding additional tidal volume may take care the dead space effect of devices but what about the conditions where already the minute volume or tidal volume requirements are high due to pulmonary disease esp severe ARDs and COvid lungs on ventilator

More significant effect on minute ventilation requirements specific to patients with severe ARDS like patients with very high VQ mismatch producing very high dead space effect .

A number of references missed.

Isoflurane vs. propofol for sedation in invasively ventilated patients with acute hypoxemic respiratory failure: an a priori hypothesis substudy of a randomized controlled trial. Becher T,. Ann Intensive Care. 2022 Dec 20;12(1):116. 

Usefulness of Inhaled Sedation in Patients With Severe ARDS Due to COVID-19. Gómez Duque M,

Capabilities of the delivery devices in the presence of very high minute ventilation requirement also need to be highlighted.

Anaesthesiol Intensive Ther  . 2022;54(1):23-29. Sevoflurane in combination with esketamine is an effective sedation regimen in COVID-19 patients enabling assisted spontaneous breathing even during prone positioning Joachim Bansbach 

Author Response

Reviewer 2 report

Point 1

Good review of inhalaltional sedation use in ICU.

There have been a number of review articles in different journals on this topic

Inhaled sedation in the intensive care unit. Jabaudon M, Zhai R, Blondonnet R, Bonda WLM. Anaesth Crit Care Pain Med. 2022 Oct;41(5):101133

Response 1:

Thank you. Yes will include the references by Jabaudon et al 2022 (ref 16). Jabaudone et al also provided our review with permission to use their figure. Reference 16

Points 2

The review only describes the advantages as feasible to give inhalational sedation and decreasing the dosages of muscle relaxants as well as sedative – which is not unexpected.

Harmful effects of prolonged inhalational agents need to be highlighted more .

Response 2:

The harmful effects of prolonged inhalation sedation was addressed and updated under the paragraphs titled: Pros and cons of prolonged inhalation sedation and What are the other Obstacles and Precautions with Inhaled Sedation Practice, as reported by several researchers.  Page 10-12

The downsides of use of volatile anesthetic agents on renal functions was discussed in a paragraph named Renal Function Under Inhaled Sedation. Page 7and 8

The effect of volatile agents on the hypoxic pulmonary vasoconstriction was added and discussed. In paragraph titled Potential effects on respiratory functions. Page 5

Air pollution risk was also mentioned as an additional harmful effect to the environment challenges of the use of volatile anesthetic for sedation. Page 8 and 9

Point 3

Effect of inhalational anaesthetic agents on hypoxic pulmonary vasoconstriction – one of the compensatory mechanism to minimise hypoxia in patients with pulmonary shunt (as happens in ARDS) also need to be highlighted.

Role of waste anaesthetic gases on environmental pollution not highlighted.

Authors have dealt with only ICU pollution and gas scavenging. The waste anaesthetic gas effect on carbon load as well as greenhouse effect needs to be highlighted. These effects are forcing anaesthetist to stop using inhalational agents in operating room for anaesthesia.

So this may not have generalised use in ICUs.

Response 3:

The effect of volatile agents on Hypoxic pulmonary vasoconstrictor id discussed and updated in the paragraph titled Potential effects on respiratory functions. Page 5

Air-pollution and global warming risk with the use of inhaled anesthetics references added and updated. References added for environmental air pollution. Page 8 and 9

Varughese, Shane, and Raza Ahmed. “Environmental and Occupational Considerations of Anesthesia: A Narrative Review and Update.” Anesthesia & Analgesia 133, no. 4 (2021): 826–35. https://doi.org/10.1213/ane.0000000000005504.

Alexander, Richard, Andrew Poznikoff, and Stephan Malherbe. “Greenhouse Gases: The Choice of Volatile Anesthetic Does Matter.” Canadian Journal of Anesthesia/Journal canadien d'anesthésie 65, no. 2 (2017): 221–22. https://doi.org/10.1007/s12630-017-1006-x. 

Point 4

Presently it may have selective role in patients requiring very high IV sedatives and also for giving gap of IV drug free time. It can be use full as supplement to high iv sedative dosages.

There is need to have more controlled studies comparing two types of sedation for more conclusive evidence.

Response 4:

Available RCT studies were added to text at various location and identified as RCT studies

RCT was pointed as one of the essential recommendations or future studies

This was added to the conclusions and future recommendations .

Point 5

Adding additional tidal volume may take care the dead space effect of devices but what about the conditions where already the minute volume or tidal volume requirements are high due to pulmonary disease esp severe ARDs and COvid lungs on ventilator

More significant effect on minute ventilation requirements specific to patients with severe ARDS like patients with very high VQ mismatch producing very high dead space effect.

Response 5: Yes we agree, this was highlighted and updated in the paragraph titled:

What are the other Obstacles and Precautions with Inhaled Sedation Practic.e Page 11 and 12

Point 6

A number of references missed.

Isoflurane vs. propofol for sedation in invasively ventilated patients with acute hypoxemic respiratory failure: an a priori hypothesis substudy of a randomized controlled trial. Becher T,. Ann Intensive Care. 2022 Dec 20;12(1):116. 

Usefulness of Inhaled Sedation in Patients With Severe ARDS Due to COVID-19. Gómez Duque M,

Capabilities of the delivery devices in the presence of very high minute ventilation requirement also need to be highlighted.

Anaesthesiol Intensive Ther  . 2022;54(1):23-29. Sevoflurane in combination with esketamine is an effective sedation regimen in COVID-19 patients enabling assisted spontaneous breathing even during prone positioning Joachim Bansbach 

Response 6:

Both references were added. Reference 75 and 76.

Capabilities of the delivery devices in the presence of very high minute ventilation requirement also need to be highlighted.

This is an important point, thank you for raising it, highlighted in page 11

This was added and discussed in paragraph titled: What are the other Obstacles and Precautions with Inhaled Sedation Practice

A new reference was added

Meiser et al reported one of the technical limitations of the AnaConda. This device has a  reflecting capacity of a minimal 10 mL of the inhalation agent in one expiration. The infusion rates of the volatile agents need to increase hand to hand with the required high increases in minute ventilation as with patients suffering from  COVID 19 to keep the end tidal concentrations of the volatile agents constant. 85

Round 2

Reviewer 1 Report

There is only the track changes version, and it becomes more difficult to read and predict an overview of the paper; what remains of many paragraphs, especially for tables.

The paper has gained clarity but still lacks proper order in the parts of the article and its content. There are still repetitions, topics mentioned in random order, and then get again. All this makes for fragmentary reading.

In the Introduction, I would write what topic the paper wants to address and its aims, but no results.

The division of topics covered in Results and Discussion is, in my opinion, incorrect.

Authors should move results and tables from the Discussion to the Results chapter.

The first sentences of the Results are summarizing and conclusive. I suggest moving them to Discussion/Conclusions.

Data recorded from the analyzed studies should match the list described in the results.

Much has been done, but there is still much to be done for the paper to be published.

Author Response

Reviewer 1

There is only the track changes version, and it becomes more difficult to read and predict an overview of the paper; what remains of many paragraphs, especially for tables.

Response

Track changes reduced and paper rearranged

The paper has gained clarity but still lacks proper order in the parts of the article and its content. There are still repetitions, topics mentioned in random order, and then get again. All this makes for fragmentary reading.

Response

Yes, Thank you. Repetitions checked and deleted as appropriate throughout the manuscript

In the Introduction, I would write what topic the paper wants to address and its aims, but no results.

Response

Introduction revised and the following paragraph moved to the results section

These features might improve the quality of recovery, as demonstrated by Ostermann et al. systematic review, Mesnil et al. clinical trial, and Blondonnet et al national survey.12-14

The division of topics covered in Results and Discussion is, in my opinion, incorrect.

Authors should move results and tables from the Discussion to the Results chapter.

Response

Thank you

All tables which represent the main results findings were moved to results section as appropriate.

Discussion of the findings were rearranged and updated to address each subtitle in more details

New (2022/2023) recent and updated references added (ref 24, 82, 85).

References renumbered according to order of appearance following the rearrangements

The first sentences of the Results are summarizing and conclusive. I suggest moving them to Discussion/Conclusions.

Response

This sentence was moved to the conclusion:

Several studies demonstrated the cardio-protective, anti-inflammatory and bronchodilator properties of inhaled anesthesia agents along with their minimal metabolism.

Data recorded from the analyzed studies should match the list described in the results.

Much has been done, but there is still much to be done for the paper to be published.

Response

All data Revised and adjusted

The manuscript was rearranged and new recent 2022/2023 references added as appropriate.

Reference 24

Jabaudon M, Zhai R, Blondonnet R, Bonda WLM. Anaesth Crit Care Pain Med. 2022 Oct;41(5):101133

Reference 28

Soukup J, Michel P, Christel A, Schittek GA, Wagner NM, Kellner P. Prolonged sedation with sevoflurane in comparison to intravenous sedation in critically ill patients - A randomized controlled trial. J Crit Care. 2023 Jan 12;74:154251. doi: 10.1016/j.jcrc.2022.154251. Epub ahead of print. PMID: 36640476.

Reference 85

Landoni G, Belloni O, Russo G, Bonaccorso A, Carà G, Jabaudon M. Inhaled Sedation for Invasively Ventilated COVID-19 Patients: A Systematic Review. J Clin Med. 2022 Apr 29;11(9):2500. doi: 10.3390/jcm11092500. PMID: 35566625; PMCID: PMC9105857.

Reviewer 2 Report

Thank you for incorporating suggestions.

Regarding Air Pollution - authors have mentioned the risk and measures for workplace pollution prevention but environmental pollution by gases which have prolonged life time may have hazardous effect on environment and if it is to be used for routine in icu  patients then its impact on environment can be huge. Though authors have mentioned the use of activated charcoal filters in expiratory limb in preventing environmental pollution, its limitation especially with high flows and high concentrations needs to be highlighted and the implications for green house gas effect explicitly defined.

Author Response

Reviewer 2

Thank you for incorporating suggestions.

Regarding Air Pollution - authors have mentioned the risk and measures for workplace pollution prevention but environmental pollution by gases which have prolonged life time may have hazardous effect on environment and if it is to be used for routine in icu patients then its impact on environment can be huge. Though authors have mentioned the use of activated charcoal filters in expiratory limb in preventing environmental pollution, its limitation especially with high flows and high concentrations needs to be highlighted and the implications for green house gas effect explicitly defined.

Response

A very important point

An additional paragraph was added and highlight in blue for this point under title

Air pollution risk

References 63-69

Reference for Air Pollution

  • Herzog-Niescery J, Seipp HM, Weber TP, Bellgardt M. Inhaled anesthetic agent sedation in the ICU and trace gas concentrations: a review. J Clin Monit Comput. 2018; 32(4): 667-675.
  1. Sackey PV, Martling CR, Nise G, Radell PJ. Ambient isoflurane pollution and isoflurane consumption during intensive care unit sedation with the Anesthetic Conserving Device. Crit Care Med. 2005; 33(3): 585-590.
  2. Accorsi A, Valenti S, Barbieri A, Raffi GB, Violante FS. Proposal for single and mixture biological exposure limits for sevoflurane and nitrous oxide at low occupational exposure levels. Int Arch Occup Environ Health. 2003; 76(2): 129-136.

66 Varughese S, Ahmed R. Environmental and occupational considerations of anesthesia: a narrative review and update. Anesthesia and Analgesia. 2021 Oct;133(4):826.

  • Alexander, Richard, Andrew Poznikoff, and Stephan Malherbe. “Greenhouse Gases: The Choice of Volatile Anesthetic Does Matter.” Canadian Journal of Anesthesia/Journal canadien d'anesthésie 65, no. 2 (2017): 221–22. https://doi.org/10.1007/s12630-017-1006-x. 
  1. Meyer MJ. Desflurane should des-appear: global and financial rationale. Anesthesia & Analgesia. 2020 Oct 1;131(4):1317-22.
  2. Herzog‐Niescery J, Vogelsang H, Gude P, Seipp HM, Uhl W, Weber TP, Bellgardt M. Environmental safety: Air pollution while using MIRUS™ for short‐term sedation in the ICU. Acta Anaesthesiologica Scandinavica. 2019 Jan;63(1):86-92.
